# Exploring the Environment behind In-Patient Falls and Their Relation to Hospital Overcrowdedness—A Register-Based Observational Study

**DOI:** 10.3390/ijerph182010742

**Published:** 2021-10-13

**Authors:** Dimitrios Stathopoulos, Eva Ekvall Hansson, Kjerstin Stigmar

**Affiliations:** Department of Health Sciences, Lund University, 22100 Lund, Sweden; distathopoulosphysio@gmail.com (D.S.); kjerstin.stigmar@med.lu.se (K.S.)

**Keywords:** falls, impatient falls, extrinsic factors, organizational factors, hospital overcrowdedness

## Abstract

(1) Background: Inpatient falls are a serious threat to patients’ safety and their extrinsic factors are, at present, insufficiently described. Additionally, hospital overcrowdedness is known for its malicious effects but its relation to the inpatient falls is currently underexplored. The aim of this study was to explore the distribution of falls and their extrinsic characteristics amongst a range of different clinics, and to explore the correlation and predictive ability of hospital overcrowding in relation to inpatient falls. (2) Methods: An observational, cross-sectional, registry-based study was conducted using retrospective data from an incidence registry of a hospital organization in Sweden during 2018. The registry provided data regarding the extrinsic factors of inpatient falls, including the clinics’ overcrowdedness. Simple descriptive statistics, correlation analysis and simple linear regression analysis were used. (3) Results: Twelve clinics were included. A total of 870 inpatient falls were registered during 2018. Overcrowdedness and total amount of falls were positively and very strongly correlated (r = 0.875, *p* < 0.001). Overcrowdedness was a significant predictor of the total amount of inpatient falls (*p* < 0.001, α = 0.05). (4) Conclusions: The characteristics regarding inpatient falls vary among the clinics. Inpatient overcrowding might have a significant role in the prevalence of inpatient falls, but further high-evidence-level studies are required.

## 1. Introduction

Falls among individuals are reported frequently all over the world and are considered a significant public health threat. It is estimated that, each year, half a million individuals are led to death from falls, with a significant proportion of them living in low- or middle-income countries [1]. Non-fatal falls are responsible for 17 million disability adjusted years lost, especially among populations of people aged 65 or above [1]. In Sweden, 5% of the injuries that require medical attention come from falls [2]. Falls are the most common reason for aged individuals to become injured and quite often have serious consequences; each year, 300,000 people seek specialist care after being injured in a fall accident [3]. The Swedish Board of Health and Welfare estimated that falls cost municipalities and county councils 1.11 billion EUR every year (primary care costs excluded) [4].

The occurrence of falls during hospitalization is a serious threat to the patient’s safety and a major issue in terms of the quality of care in many hospitals and health institutions around the world [5]. Falls of patients during their hospital care degrade the environmental safety of the organization as they represent a lack of security and an absence of protection, and are often a cause of an increase in hospitalization days for the individual, as well as the worsening of their recovery [6]. Inpatient falls constitute two out of five unfortunate events in hospitals and their frequency varies from 2 to 12 for every 1000 patients each day [7]. Oftentimes, they result in trauma, the accidental removal of peripheral medical equipment and fear of falling for the inpatient. The generated fear of falling has been shown to lead to emotional changes and decreased mobility, resulting in muscle weakness, contracture, thrombogenic events, postural hypotension, general clinical worsening and even death [6,8].

The incidence of inpatient falls varies depending on each country. In 2007, approximately one million of patient falls were reported during hospitalization in the USA, while during 2008, 280,000 falls of inpatients were reported in the U.K. [8]. There are no official estimates regarding inpatient falls in Sweden, but it is worth mentioning that one out of three hip fractures occur, in the Swedish hospitalized population, following a fall [9]. We should also take into consideration that the above speculations may be higher because a significant proportion of inpatient falls remains unreported [10].

Inevitably, falls during hospitalization have a great impact in terms of increases in healthcare costs. It is estimated that patient falls in hospitals cost approximately 100 million EUR per year in the U.K., while in the U.S.A., an accidental fall in the hospital will cost the patient 3.8 thousand EUR more due to the extension of their hospitalization [11,12]. Finally, falls during hospitalization not only affect the patients’ well-being and expenses but they are also very likely to cause anxiety and guilt among the hospital’s staff due to the feeling that someone is responsible or that measures could have been taken to prevent the fall. Moreover, complaints have been made, and even litigation has been undertaken, by patients’ families or caregivers [13,14]. Hospitalized individuals are described as having a high risk of experiencing a fall because of the unfamiliar hospital environment and the adverse clinical situation in which they find themselves [15].

In order for a health care system to understand and prevent inpatient falls, it is crucial to retain more knowledge on factors that cause or are associated with the phenomenon. The literature suggests two large categories of factors that are related to inpatient falls: intrinsic factors and extrinsic factors [16]. Intrinsic factors consist of characteristics that describe the patient’s health and overall condition, and extrinsic factors are characteristics that describe the environment and the setting under which the fall occurred.

There is a plethora of published studies describing the intrinsic factors for inpatient falls [5,12,13,14,15,16,17,18,19,20,21,22,23,24,25,26,27,28]. The strongest predictor seems to be a previous history of falling, especially amongst elderly individuals. The intrinsic factors have been studied in the context of different patient and health care center populations in a such detailed way that two new subcategories of intrinsic factors were proposed, and are currently under investigation, in addition to the physical-related factors: psychological and socio-economic factors [27]. On the other hand, due to the considerable variety in terms of the clinical environment and organizational level of each health center around the world, the extrinsic factors that contribute to the patients’ falls are not yet fully determined.

The term “organizational level” refers to all major functions and services provided by the hospital or the health center, such as the safety measures, the amount of specialized personnel and their responsibilities, and so on. Some well-known extrinsic fall-related factors, according to the literature, are wet floors, the locations of bathrooms, low nurse-to-patient ratios, and the geriatric wards of hospitals [8]. There are many more extrinsic factors yet to be explored and determined that are unique to each health center. Hence, there is a significant need for more extensive studies on the topic since extrinsic factors are highly related to the prevalence of falls [7,8,29]. In addition to the previous extrinsic factors, patient overcrowding of a hospital setting could be a considerable extrinsic factor for inpatient falls. Generally, a health care center is considered overcrowded when the identified need for services exceeds the available resources for patient care in the health center [30]. Regarding Sweden, the Swedish Association of Local Authorities and Regions define “hospital overcrowdedness” as a situation in which an enrolled patient is cared for in a health care facility that does not meet the requirements regarding the availability of care due to resources being unavailable (e.g., a patient with a neurological disorder is hospitalized in a non-neurological ward because the neurological ward is full) [31]. On top of that, among the European countries, Sweden has the second lowest number of hospital beds per hundred thousand inhabitants [32], but a very high number of practicing medical personnel per 100,000 inhabitants [33].

Previous studies regarding the effects of overcrowding in hospitals have shown that when a health care center is overcrowded by patients, the caregiver-to-patient ratio decreases and, consequently, adverse events are more likely to occur [34,35,36,37]. Inpatient overcrowding is related to a decreased quality of services received by inpatients, and even the death of inpatients due to medical errors [34]. An overcrowded hospital setting seems to be confusing and disorients the health care personnel since it exceeds the system’s capacity, causing vital support services to be overwhelmed and, thus, to become inadequate in terms of keeping up with the patients’ needs. Medical personnel’s perceptions regarding overcrowded hospital wards reveal how malicious that kind of hospital setting can be for them and the patients; lack of workspace, elusive care and loss of authority are some of the commonly reported feelings from nurses that usually work in overcrowded departments. Inevitably, this kind of environment leads to personnel safety issues, as well as moral distress and stress-related burnout among the medical personnel [38,39,40].

Inpatient falls and hospital overcrowdedness could be highly correlated to each other, and thus, studies are required to further explore this relationship; an in-depth literature search for evidence of the relation between overcrowdedness and inpatient falls did not show a sufficient amount on this topic. To the best of our knowledge, there is only one study that examines patient overcrowding in relation to the risk of falls: in 2016, Teitelbaum and colleagues, via a retrospective registry-based study, concluded that patient overcrowding was highly associated with the occurrence of falls in a psychiatric ward [41]. The overall aim of this study was to survey and explore the inpatient falls and their extrinsic/organizational factors within a hospital organization during 2018. Furthermore, the relation between overcrowdedness and inpatient falls was studied.

## 2. Materials and Methods

This is a register-based observational cross-sectional study that uses retrospective data from an incidence registry of falls. We collected data from a register of inpatient falls at a large University hospital organization in South Sweden. Skåne University Health Care includes two University hospitals and three smaller hospitals that are all located in the southwest corner of Skåne in the south of Sweden. It covers a wide range of health care specialties including a total of eighteen clinics.

The organization maintains a detailed registry system where all incidences of inpatient falls are reported, registered, and saved to a database. For each reported fall, pre-determined information is registered. According to the organization’s personnel, at least 90–95% of the inpatient falls that occur are reported to the registry. Based on the registry, for each clinic of the organization, there is information about: the total amount of inpatient falls, and the total amount of severe and non-severe falls (where a severe fall is a fall that leads to a serious injury such as a fracture, a concussion or death, and a non- severe fall is a fall that leads to minor injuries such as a bruise, light pain or no adverse outcome); the times and dates of falls; the total amount of falls that happened in the room, toilet, corridor, and dayroom of the clinic; the total amount of falls that happened while health care personnel were present or not present; and number of patients exceeding the maximum number of patients at each clinic (described as overcrowdedness in this study).

The registry provides information concerning all of the clinics within the Skåne University Health Care organization. The chosen time interval of the examined clinics, regarding their inpatient falls, was between 1 January 2018 and 31 December 2018. The inclusion criterion for those clinics was that they should be inpatient clinics, and the exclusion criterion was that the clinics should not have any missing data regarding overcrowdedness and/or the characteristics of their inpatient falls.

For the statistical description and analysis, IBM SPSS Statistics version 26 [42] was used. Data were imported to SPSS and thirteen variables were created. There was one categorical nominal string variable, which described the name of the included clinics, and twelve numeric variables describing, for each clinic, the total amount of: inpatient falls, severe falls, non-severe falls, falls in the toilet, falls in the room, falls in the corridor, falls in the dayroom, falls during the day, falls during the night, falls where staff were present, falls where staff were not present, and overcrowdedness. The overcrowdedness of each clinic was presented in the registry as the number of patients that exceeded the maximum capacity of each clinic at a given time, and could be presented on a daily, weekly, and monthly basis. In this study, we were interested in a yearly measurement of overcrowdedness for each included clinic; thus, we summed up all the monthly values of each clinic and divided them by 12. Consequently, we established the mean number of patients exceeding the maximum capacity for each clinic during 2018, and we named the variable overcrowdedness. For the purpose of this study, we suggested that an overcrowdedness value of 1 or more indicates an overcrowded clinic. In order to explore the correlation between the inpatient overcrowdedness of each clinic and the total amount of inpatient falls of each clinic, the Pearson’s correlation coefficient was used since parametric testing was possible from the given data [43]. For the interpretation of the correlation, we suggested that a correlation coefficient of less than 0.3 is poor, 0.3 to 0.5 is fair, 0.6 to 0.8 is moderately strong and at least 0.8 is very strong [44]. In order to examine whether the inpatient overcrowdedness of a clinic can statistically explain the total amount of inpatient falls in the clinic, a simple linear regression analysis was used [45]. For all of the statistical tests, the significance level α was set to 0.05. All of the chosen statistical tests were analyzed in terms of their assumptions [46] in order to obtain the most robust results possible.

The study was approved by the Ethical Review Board, Dnr 2019-05300.

## 3. Results

Twelve out of the eighteen clinics were accepted for the study. Two clinics were not inpatient clinics and four clinics had missing data regarding their overcrowdedness. The included clinics were the following: the Endocrinology clinic, the Gastroenterology clinic, the Infectious Diseases clinic, the Vascular clinic, the Gynecology clinic, the Lung clinic, the Neurology clinic, the Nephrology clinic, the Neurosurgery clinic, the Orthopedic clinic, the Hematology clinic and the Urology clinic.

### 3.1. Inpatient Falls and Their Characteristics

During the one-year period, there were, in total, 870 inpatient falls registered at the included clinics. Figure 1 shows how these were distributed among the 12 included clinics during 2018.

Regarding the severity of those inpatient falls, 35 out of 870 were described as severe, leading to a serious injury or death, while 835 out of 870 were described as non-severe, leading to minor injuries or no outcomes. Falls occurred almost equally in the daytime or at night: 416 occurred during the daytime (06:00–18:00) and 464 occurred during the night (18:00–06:00). Most of the falls took place in the rooms of patients or in toilets. Only in a minority of cases (*n* = 93) were hospital staff present when the fall happened. The extrinsic characteristics of inpatient falls are presented in Table 1.

### 3.2. Correlation between the Inpatient Overcrowdedness and the Total Amount of Inpatient Falls

The collected data was consistent with the assumptions [46] given for the parametric testing of the correlation between the overcrowdedness variable for each clinic and total amount of falls for each clinic; consequently, the Pearson’s correlation coefficient was used. The total amount of inpatient falls in each clinic was strongly associated with the overcrowdedness of each clinic (r = 0.875, *p* < 0.001). A positive and very strong correlation was found between the two variables.

### 3.3. Statistical Association of Total Number of Inpatient Falls with the Inpatient Overcrowdedness

A significant degree of regression was found (F1 = 32.515, *p* < 0.001), with an R2 of 0.765, using the equation given below. Patient overcrowdedness was a significant predictor of the total amount of inpatient falls (*p* < 0.001):y = −8.7 + 53.42 × x(1)

According to the model, the total amount of inpatient falls in each clinic increases by approximately 53 for every point that is added to the mean number of patients over the maximum capacity of a clinic.

## 4. Discussion

In this study, we found that inpatient falls are more frequent at the orthopedic, neurology and infectious disease clinic. Furthermore, we found a positive and strong correlation between inpatient falls and overcrowdedness. Further analyses suggest a rate of increase of 53 inpatient falls per year, if the mean number of patients exceeding the maximum capacity of each clinic per month, over the course of one year, is more than one.

A total of 870 inpatient falls were reported among the twelve clinics. We cannot compare or interpret this total as being high or low since there is no cut-off point available in the literature for the total number of falls in a hospital setting. The only reference point that is available is that inpatient falls generally happen 2.3 to 7 times per 1000 patient days [47], but the gathered data from this study did not allow for the calculation of patient days. The neurology and orthopedic clinic had the largest amounts of inpatient falls, constituting the 40% of the total amount of inpatient falls within the organization. These findings were expected since patients in such clinics usually have functional limitations and activity restrictions that constitute significant intrinsic factors for a fall [6,8,12]. The findings are in line with previous studies regarding the epidemiology of inpatient falls [7,48].

It is worth noting the high number of inpatient falls in the infectious disease clinic. At first glance, it is hard to explain why a clinic of this type presents a significant number of inpatient falls, since patients, in general, are not expected to have intrinsic factors related to falls. Extrinsic or organizational factors could explain the high number of inpatient falls in the clinic, such as the high overcrowdedness (a mean of 1.9 patients above the maximum capacity during 2018), but a further in-depth investigation is required. Between the rest of the examined clinics, there was no significant variance observed in the total number of inpatient falls.

During 2018, 35 severe inpatient falls were registered. We cannot determine if this is a high number or not. In a similar study, Hitcho et al. explored the circumstances of inpatient falls in a hospital setting during a period of four months in which four severe inpatient falls were recorded [7]. In our study, the vascular clinic had 21% of falls classified as severe, which was much higher than the other clinics.

Inpatient falls in the organization did not seem to follow a specific pattern during the day. It has previously been suggested that inpatient falls are more likely to happen during the daytime, as the patients are most active at this time, but these findings vary between each study [6]. Other suggested high-risk time periods for inpatient falls are the early morning and late afternoon periods where the personnel shift change is taking place [6]. All these high-risk periods may differ between hospitals due to different organizational plans and constructs.

The findings showed that the vast majority of the inpatient falls took place in the patients’ rooms, and this was observed at all of the included clinics. Along with the toilet falls (since toilets are located in the patients’ rooms), 89% of the registered inpatient falls happened there. Similar studies have found the patient’s room and the toilet to be very high-risk areas for a fall for multiple reasons [5,7,49]. The findings from this study verify the hypothesis of previous studies that the location of the patient’s room is considered a significant extrinsic inpatient fall factor [8].

A very high percentage of inpatient falls (89%) happened while the medical personnel were absent. This finding is in line with the literature as it has been previously observed that inpatients tend to fall while they are unattended by medical personnel because they become involved in actions of which they are not aware of the difficulty or danger [50]. In this study, we found that overcrowdedness was associated with inpatient falls. The most likely assumption is that overcrowdedness resulted in a lack of staff being present.

Eight out of the twelve examined clinics were identified as overcrowded during 2018. The neurology and orthopedic clinics were the most overcrowded places, with values of 3.5 and 3, respectively, representing the mean number of patients exceeding the maximum capacity each month. The fact that 75% of the clinics within the organization were overcrowded by inpatients further highlights the previously reported low number of hospital beds per hundred thousand inhabitants in Sweden [32]. While these statistical speculations exclusively concern the examined hospital organization and, in no cases, reveal a causation between inpatient falls and overcrowding, they lay the foundations for further in-depth scientific investigation on the subject, which is urgently needed [41]. The malicious effects of overcrowding in the clinics have been observed in a plethora of studies, from both the patients’ and medical personnel’s perspectives [41]. Since overcrowdedness of clinics is proven to cause burnout and dissatisfaction in medical personnel [51], negative outcomes and discomfort in patients, and a reduction in their safety during hospitalization [52], inpatient falls could occur more frequently. Additionally, it was previously reported that adverse events, such as falls, are more likely to happen during periods of increased bed occupancy, although empirical data in this field are rare in the literature [41]. Teitelbaum et al., in their unique study concerning patient overcrowding and inpatient falls, suggested that patient overcrowding in a hospital setting should be added to the other extrinsic risk factors for inpatient falls [41].

The main strength of this study is the register-based design, where we had access to data from a large hospital organization for one year. Although the data were reported by different health professionals at the various clinics, the coverage for the register was considered to be very good. In a similar (but longitudinal) observational study conducted by Abreu et al. [5], the authors identified a decrease in inpatient falls after the start of the study and the use of an incidence registry because the hospital personnel were aware of the project and tried to avoid negative results. In addition, another positive aspect of this study is that it used an incidence registry with previously gathered data and made valuable use of them without the data presenting an obstacle to the organization’s everyday operations and without using any personal information from the inpatients of the organization.

Another strength is the exploration of patient overcrowding and the connection of this phenomenon to the total number of inpatient falls, an aspect that was absent from previous studies regarding the exploration of organizational factors that contribute to inpatient falls [41]. The linear regression model quantified the overcrowdedness phenomenon as the inpatient fall count, which is of great importance for the organization. Additionally, the gathered data and their statistical properties permitted the use of parametric statistical analysis, which generally provides trustworthy results.

Due to the complexity of the gathered data, we created a variable based on the registry’s information regarding the clinics’ overcrowdedness, which indicates the mean number of patients over the maximum capacity of each clinic during the examined time period. The cut-off point for a clinic to be identified as overcrowded was set as equal to or greater than 1 mean patient, which is a relatively “tolerant” definition for overcrowding since the optimal occupancy for a clinic, in order for it to function properly, was previously reported to be a 85% of the maximum bed occupancy [41].

On the other hand, there are some noteworthy limitations that characterize the study. The retrospective aspect of the study and the lack of a control group does not allow the determination of cause-and-effect relationship between hospital overcrowdedness and inpatient falls, so the positive correlation must be interpreted with cautiousness and can give guidance for future studies. The absence of the medical and geriatric clinics from the study due to missing data regarding their overcrowdedness indicator constituted a significant absence in this study. Generally, those clinics tend to have many inpatient falls and their inclusion would have provided a very reliable source of coverage regarding inpatient falls. Moreover, the geriatric clinic is a place where, according to the literature, most of the inpatient falls occur; thus, the exploration of the environmental factors of those inpatient falls would have provided a better understanding of the phenomenon in the organization. Regarding the data from the incidence registry, there was a transformation of the overcrowdedness for each clinic because the registry provided daily, weekly and monthly information related to the number of patients exceeding the maximum capacity of each clinic, and we were interested in obtaining a yearly measurement. This transformation of a pure value into a mean value of inpatients could have skewed (positively or negatively) the overcrowdedness of some clinics, leading to potentially unreliable results. The results of the study presented a very strong correlation between the number of inpatient falls and the organization’s overcrowdedness, but these findings should be interpreted with caution; inpatient falls have a huge variety of extrinsic factors that were not covered in this study, such as the patient-to-nurse ratio of each clinic, the fall prevention measures/interventions/equipment of each clinic, etc. We should also keep in mind that since inpatient falls tend to happen 2 to 12 times per 1000 patient days [7], it is expected that a large number of inpatient falls would inevitably occur in a crowded hospital, without the phenomenon of overcrowdedness being the primary factor.

## 5. Conclusions

In this study, we found that the majority of inpatient falls within the included clinics happened at the Orthopedic, Neurology and Infectious disease clinics. Eight out of twelve clinics were overcrowded during 2018 and overcrowdedness was positively and strongly associated with the number of falls. The analyses suggest that an increase of approximately 53 inpatient falls occurred in one-year, during which the increase in the mean number of patients that exceeded the maximum capacity of each clinic per month, over the course of one year, was at least one.

## Figures and Tables

**Figure 1 ijerph-18-10742-f001:**
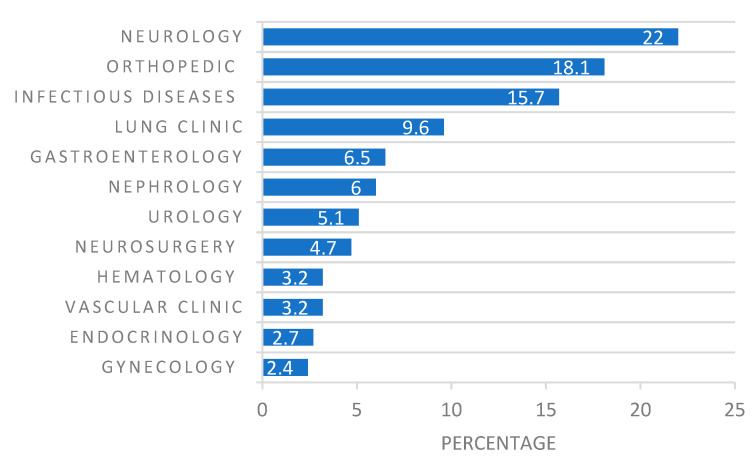
Distribution of inpatient falls (percentage) among included clinics in relation to the total number of falls (*n* = 870) during 2018.

**Figure 2 ijerph-18-10742-f002:**
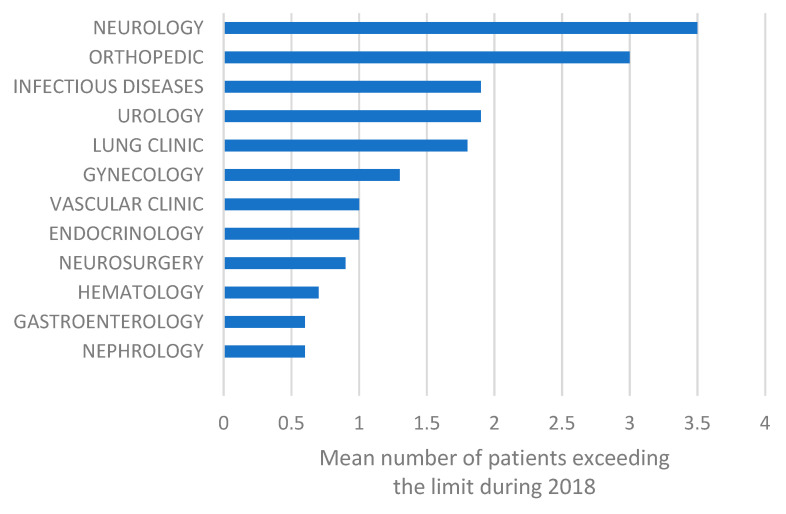
Mean value of patient overcrowdedness at the different clinics. A value of ≥1 indicates overcrowdedness.

**Table 1 ijerph-18-10742-t001:** Inpatient falls at the different clinics (*n* = 12) and extrinsic characteristics of the inpatient falls.

Clinic	Falls	Severe	Location				Time of Day		Staff
		Falls	Toilet	Dayroom	Room	Corridor	Day	Night	Present
	*n*	*n* (%)	*n* (%)	*n* (%)	*n* (%)	*n* (%)	*n* (%)	*n* (%)	*n* (%)
Gynecology	21	1 (4.8)	11 (52.4)	1 (4.8)	7 (33.3)	2 (9.5)	12 (57.1)	9 (42.9)	3 (14.3)
Endocrinology	24	2 (8.3)	6 (25)	0 (0)	18 (75)	0 (0)	9 (37.5)	15 (62.5)	1 (4.2)
Vascular	28	6 (21.4)	2 (7.1)	2 (7.1)	22 (78.6)	2 (7.1)	13 (46.4)	15 (53.6)	3 (10.7)
Hematology	33	3 (9.1)	8 (24.2)	1 (3)	23 (69.8)	1 (3)	13 (39.4)	20 (60.6)	5 (15.2)
Neurosurgery	41	1 (2.4)	7 (17.1)	1 (2.4)	28 (68.3)	5 (12.2)	24 (58.5)	17 (41.5)	4 (9.8)
Urology	45	1 (2.2)	8 (17.8)	3 (6.7)	30 (66.7)	4 (8.9)	28 (62.2)	17 (37.8)	4 (8.9)
Nephrology	52	3 (5.8)	9 (17.3)	6 (11.5)	34 (65.,4)	3 (5.8)	24 (46.2)	28 (53.8)	5 (9.6)
Gastroenterology	57	1 (1.8)	20 (35.1)	0 (0)	35 (61.4)	2 (3.5)	30 (52.6)	27 (47.4)	11 (19.3)
Lung	84	3 (3.6)	20 (23.8)	3 (3.6)	58 (69)	3 (3.6)	31 (36.9)	53 (63.1)	7 (8.3)
Infectious diseases	137	3 (2.2)	34 (24.8)	1 (0.7)	96 (70.1)	6 (4.4)	64 (46.7)	73 (53.3)	13 (9.5)
Orthopedic	158	5 (3.2)	28 (17.7)	8 (5.1)	116 (73.4)	6 (3.8)	68 (43)	90 (57)	11 (7)
Neurology	190	6 (3.2)	32 (16.8)	12 (6.3)	123 (64.7)	23 (12.1)	90 (47.4)	100 (52.6)	26 (13.7)
TOTAL	870	35 (4.2)	185 (21.3)	38 (4.4)	590 (67.8)	57 (6.5)	406 (46.7)	464 (53.3)	93 (10.7)

Overcrowdedness was not equally distributed among the included clinics (Figure 2). Only four clinics had overcrowdedness values below one (1), and these were the nephrology, hematology, neurosurgery and gastroenterology clinics.

## Data Availability

The data that support the findings of this study are available from Skånes Universitetssjukhus but restrictions apply to the availability of these data, which were used under license for the current study, and thus, are not publicly available. Data are, however, available from the authors upon reasonable request and with permission of Skånes Universitetssjukhus.

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
