# Peer review of "Exploring the Environment behind In-Patient Falls and Their Relation to Hospital Overcrowdedness—A Register-Based Observational Study"

_ijerph, 2021, doi:10.3390/ijerph182010742_

Round 1
Reviewer 1 Report
The paper presents the important issue of patient safety during hospitalization and precisely the case of inpatient falls. It is well described and presenting interesting data, but I have several concerns and suggestions for the authors:
I would reconsider the title of the paper. It is too long.
Methods:
I have concerns regarding your definition of overcrowdedness. First, you should describe the research environment - Skåne University Health Care – in terms of staff capacity and crowdedness. It may be more apparent if you divide crowdedness into categories: high, mid, low, and analyze data according to levels of crowdedness. One patient over the maximum capacity is not the same as six patients over the maximum capacity. There is also a question of the capacity of the medical staff.
Results:
Please add percentages in table 1.
I suggest adding statistical tests examining differences between Day/Night, differences between staff present/not present, and differences among fall places.
I suggest removing the regression equation and figure 4 and instead add a table presenting the model predictors, B, SEB and Beta, R2 and F.
Discussion:
It is necessary to shorten the discussion and avoid repetitions.
Reviewer 2 Report
Dear Authors,
Many thanks for the opportunity to review the paper. I find the paper interesting but i feel there are further room for improvement as detailed below.
i. Line 129-131 should be moved and form last sentence in section 1.
ii Line 23 (abstracts) reported 12 clinics considered in the study, however there was mention of 18 clinics mentioned in line 137 (Materials and Methods). Authors need to reconcile this moving forward.
iv. it will be mus better if data reported from figure 1 (line 200-203) are reported in percentage and number in bracket compared to the present format adopted.
iv. It there was limited description offered to figure 2 content. In its present it made the figure redundant. Advised authors to revise.
v. Authors should endeavour to guide readers by describing key findings contained in each table/figure to which there was no comment offered to guide understanding of the table 1 content.
vi. While the essence of the discussion section is to discussion is to interpret and describe the significance of study's findings in light of what was already known about the research problem being investigated and to offer explanation around any new understanding or insights that emerged as a result of the study been considered, in this case i find the content needing second look to help achieve this goal as in some instance it was more than descriptive in nature.
vii. It will be good for the authors to consider limitation to the study/approach in their submission as well.
viii. The conclusion is weak and does not help to offer directional guide around the impact intended to achieve.
Best regards
Round 2
Reviewer 2 Report
Dear Authors,
Many thanks for the effort made at improving the paper further. I have gone through the paper and now satisfied with the updated version.
Best wishes